# Confusing Onset of MOGAD in the Form of Focal Seizures

**DOI:** 10.3390/neurolint17030037

**Published:** 2025-02-27

**Authors:** Małgorzata Jączak-Goździak, Barbara Steinborn

**Affiliations:** Department of Developmental Neurology, Poznan University of Medical Sciences, 61-701 Poznań, Poland

**Keywords:** MOGAD, focal seizures, anti-MOG antibodies, recurrent MOGAD

## Abstract

MOGAD is a demyelinating syndrome with the presence of antibodies against myelin oligodendrocyte glycoprotein, which is, next to multiple sclerosis and the neuromyelitis optica spectrum, one of the manifestations of the demyelinating process, more common in the pediatric population. MOGAD can take a variety of clinical forms: acute disseminated encephalomyelitis (ADEM), retrobulbar optic neuritis, often binocular (ON), transverse myelitis (TM), or NMOSD-like course (neuromyelitis optica spectrum disorders), less often encephalopathy. The course may be monophasic (40–50%) or polyphasic (50–60%), especially with persistently positive anti-MOG antibodies. Very rarely, the first manifestation of the disease, preceding the typical symptoms of MOGAD by 8 to 48 months, is focal seizures with secondary generalization, without typical demyelinating changes on MRI of the head. The paper presents a case of a 17-year-old patient whose first symptoms of MOGAD were focal epileptic seizures in the form of turning the head to the right with the elevation of the left upper limb and salivation. Seizures occurred after surgical excision of a tumor of the right adrenal gland (ganglioneuroblastoma). Then, despite a normal MRI of the head and the exclusion of onconeural antibodies in the serum and cerebrospinal fluid after intravenous treatment, a paraneoplastic syndrome was suspected. After intravenous steroid treatment and immunoglobulins, eight plasmapheresis treatments, and the initiation of antiepileptic treatment, the seizures disappeared, and no other neurological symptoms occurred for nine months. Only subsequent relapses of the disease with typical radiological and clinical picture (ADEM, MDEM, recurrent ON) allowed for proper diagnosis and treatment of the patient both during relapses and by initiating supportive treatment. The patient’s case allows us to analyze the multi-phase, clinically diverse course of MOGAD and, above all, indicates the need to expand the diagnosis of epilepsy towards demyelinating diseases: determination of anti-MOG and anti-AQP4 antibodies.

## 1. Introduction

MOGAD (myelin oligodendrocyte glycoprotein antibody disease) is a demyelinating syndrome characterized by antibodies against myelin oligodendrocyte glycoprotein (MOG-IgG). Apart from multiple sclerosis and the spectrum of neuromyelitis optica, it is one of the manifestations of the demyelinating process, more common in the pediatric population [1,2,3,4]. MOGAD may take a variety of clinical forms: acute disseminated encephalomyelitis (ADEM), retrobulbar optic neuritis (ON), often binocular ON, transverse myelitis (T.M.), or NMOSD-like course (neuromyelitis optica spectrum disorder-like phenotype) [1,2,5,6,7,8]. The course may be monophasic (40–50%) or multiphasic (50–60%), especially in the case of persistently positive titers of anti-MOG antibodies [1,2,7]. Focal seizures with secondary generalization [2,9], without typical demyelinating lesions in brain MRI and CSF abnormalities [10], constitute infrequent first manifestations of the disease, preceding the typical symptoms of MOGAD by 8 to 48 months. Another uncommon presentation of MOGAD is aseptic meningitis, characterized by leptomeningeal enhancement on magnetic resonance imaging (MRI) and a prolonged fever lasting more than seven days as the primary manifestation [11,12,13,14,15,16]. The article presents a case of a teenager whose first symptoms of MOGAD included focal motor epileptic seizures and preceded full-blown MOGAD by several months.

## 2. Case Report

A 17-year-old patient was admitted to the neurology ward in December 2020 due to seizure episodes, which involved turning the head to the right with the elevation of the left upper limb and salivation. The seizures occurred after the surgical removal of the right adrenal tumor (ganglioneuroblastoma) 5 months earlier. Then, despite a normal brain MRI result and a typical result of cerebrospinal fluid examination, the absence of onconeural antibodies in the serum and in the cerebrospinal fluid, paraneoplastic syndrome was suspected. In the oncology ward, the patient was administered intravenous steroids (methylprednisolone 1.0 g IV for five days) and immunoglobulins (a total dose of 2.0 g IV). Moreover, he underwent eight plasmapheresis procedures, and valproic acid was included in the treatment at a target dose of 20 mg/kg of body weight. However, four generalized episodes of tonic–clonic seizures were also observed during treatment.

During hospitalization in the neurology ward, Holter EEG was performed in which seizure episodes corresponded to changes in the recording. Interictal EEG showed rapid activity followed by slow theta waves and several sharp and slow wave complexes (Figure 1 and Figure 2). Topiramate was added to valproic acid at a target dose of 2 mg/kg of body weight. Gradual seizure relief was observed in the following weeks, and there were no other neurological manifestations for another nine months. After that time, in December 2021, the patient was admitted to the pediatric ward due to impaired consciousness and persistent vomiting, followed by paresis of the lower limbs. A brain MRI was performed, which showed “T2/FLAIR images reveal poorly defined areas of increased signal in the brain’s white matter, both above and below the tentorium, as well as in the periventricular and subcortical regions. These areas correspond to low signal regions in T1 images. The largest change area measures 19 × 14 mm and is located in the right middle cerebellar peduncle, which merges with changes observed in the pons. Additionally, there is a lesion in the right thalamus, measuring 10 × 24 mm, along with numerous scattered foci in the corona radiata and between the basal nuclei. Overall, the imaging findings primarily suggest acute disseminated encephalomyelitis (ADEM)” (Figure 3 and Figure 4). Cerebrospinal fluid examination revealed oligoclonal bands and an elevated protein level of 118.5 mg/dL (reference range: 15.0–45.0 mg/dL) and a slightly elevated leukocyte level of 29/mm^3^ (lymphocytes 72.4%, neutrophils 27.6%).

Additionally, a clear titer of 1:100 for anti-MOG antibodies was detected in the serum. The patient received steroids (methylprednisolone 1.0 g IV for five consecutive days). Despite initial improvement, pathological drowsiness was observed several days later, vomiting and fever recurred, and the patient was unable to walk unaided. Follow-up T2-weighted/FLAIR brain MRI revealed a new focus of an increased signal in the left thalamus.

ADEM recurrence was recognized. The patient received immunoglobulins (2.0 g IV). After rehabilitation, an improvement in the neurological condition was observed over the next four weeks. The boy returned to school. Subsequently, he reported visual acuity disturbances in the left eye. Therefore, he was hospitalized in the neurology department. On admission, neurological examination showed horizontal nystagmus, lack of abduction in the right eye, dysarthric speech, and cerebellar syndrome. Brain MRI revealed numerous foci of increased signal on T2-weighted and FLAIR sequences, with the largest one occurring in the right cerebellar hemisphere accompanied by the involvement of the peduncle (Figure 5 and Figure 6) and bilateral involvement of the dentate nuclei of the cerebellum. Foci of increased signals on T2-weighted and FLAIR sequences were also found on MRI of the cervical and thoracic spine, meeting the LETM (longitudinally extensive transverse myelitis) criteria (Figure 7). Moreover, an orbital MRI showed “the left optic nerve was thickened to 3–4 mm, with increased T2 signal intensity indicating inflammation; the optic chiasm and right optic nerve were normal” (Figure 8). Again, positive serum anti-MOG antibodies were found in a clear titer of 1:100, while anti-AQP4 antibodies tested negative. Recurrent MOGAD was diagnosed based on the overall picture, and immunoglobulins were administered (a total dose of 2.0 g IV for five days). Oral azathioprine was included in the chronic treatment at a dose of 2 mg/kg body weight and oral methylprednisolone at a dose of 1 mg/kg body weight. After ten weeks (May 2022), right optic neuritis occurred during an attempt at withdrawing glucocorticosteroids. Methylprednisolone was reintroduced at a dose of 5 × 1.0 g intravenously. An MRI of the head revealed two new subcortical foci: one located in the right frontal lobe measuring 13 × 7 mm and another in the left parietal lobe measuring 19 × 14 mm (Figure 9). Additionally, the right optic nerve exhibited blurred outlines (Figure 10) and an increased signal in the central area, along with a marginally enhancing focus in the middle part of the right optic nerve measuring 2 mm in diameter. An ophthalmological consultation was conducted, revealing color vision deficiencies in the right eye across all axes, with no reading capability in the left eye. An OCT examination showed that the retinal nerve fiber layer (RNFL) in the right eye was normal (higher than a month earlier, suggesting a possible inflammatory process), while the left eye exhibited significantly reduced RNFL and ganglion cell count. Atrophy of the left optic nerve was diagnosed. Based on the clinical presentation, recurrent optic neuritis was diagnosed as a manifestation of another MOGAG, correlating with the onset of right optic neuritis following the discontinuation of methylprednisolone. Therefore, a decision was made to increase the dose of azathioprine to 2.5 mg/kg body weight and to maintain methylprednisolone for the following weeks. After the diagnosis of MOGAD with a typical radiological and clinical picture (ADEM, MDEM, recurrent ON) and the initiation of chronic immunosuppressive treatment, no more epileptic seizures were observed. It was decided to discontinue antiepileptic drugs, starting with gradual valproic acid.

## 3. Discussion

According to the current criteria, three elements are necessary to diagnose MOGAD: 1—typical clinical picture of the demyelinating syndrome: ADEM, ON, TM or, less frequently, cerebral cortical encephalitis, brainstem presentation or cerebellar presentation; 2—the presence of anti-MOG antibodies in a clear titer; 3—exclusion of another/better diagnosis including MS [17,18,19,20,21,22]. If anti-MOG antibodies are present in low titers and negative in serum but positive in the cerebrospinal fluid, then, in addition to the clinical picture, additional supporting radiological criteria for specific demyelinating syndromes are required [21] (Table 1):

Considering the criteria mentioned above, we focused on two demyelinating syndromes for differential diagnosis: multiple sclerosis and neuromyelitis optica spectrum disorder (NMOSD). To accurately diagnose demyelinating lesions, we conducted tests to detect antibodies against aquaporin-4 (AQP4) and myelin oligodendrocyte glycoprotein (anti-MOG), following the established gold standard [9]. In addition to the positive/clear titer of anti-MOG antibodies and the exclusion of antibodies against AQP4, imaging tests—resonance imaging (MRI) of the head and spinal cord—help make the diagnosis. Unlike MS, magnetic resonance imaging of the brain reveals lesions with unclear boundaries, more often located subcortically, with a characteristic cloud-like shape and a particular location in the cerebellar peduncles [1,5,10,23]. MOGAD lesions are extensive, often bilateral, and located in the thalamus and basal ganglia [1,23]. In NMOSD, brain lesions are more often located near the third ventricle [5]; the involvement of the area postrema is particularly characteristic, and the lesions usually surround the lateral ventricles [5,6].

Unlike MS, but similarly to NMOSD, in MOGAD, the demyelinating lesions in the MRI of the spinal cord are longitudinal [2], they involve more than three segments, edema is often observed, the foci are located in the central part of the spinal cord, and they produce hypointense changes in T1 images magnetic resonance imaging [1,5,23]. Demyelinating foci in the brain and spinal cord in the course of MOGAD disappear in 50–80% of cases after the relapse [1,5,10]. MRI of the orbits in MOGAD more often than in MS describes extensive involvement of the optic nerve, covering more than half of the length of the nerve [5] before the optic chiasm with contrast enhancement around the nerve [1,23] and inflammation of the soft tissues of the orbit in up to 50% of patients with ON [2]. In optic neuritis in the course of NMOSD, the optic nerve is affected mainly in the chiasm area, with involvement of the optic pathway [3] and, unlike ON, in the course of MOGAD, there is no involvement of the soft tissues of the orbit [2]. Involvement of the optic nerves in the course of MOGAD in approximately 50% may be bilateral at the same time [1,2,5,24].

Another test supporting the proper diagnosis of demyelinating syndrome is OCT RNFL. In this study, patients with MOGAD had more involved retinal fibers compared to patients with retrobulbar optic neuritis due to MS. Despite the rapid improvement of vision in MOGAD, OCT examination reveals advanced optic nerve atrophy [1,2,10,24]. 

Examination of cerebrospinal fluid is also an auxiliary test; it allows the exclusion of infectious conditions, but the presence of oligoclonal bands does not exclude the diagnosis of MOGAD; in 15–30% of MOGAD patients, oligoclonal bands are positive [1,23]. In the examination of cerebrospinal fluid in patients with MOGAD, the level of white blood cells may be increased in 50% of cases (>5 WBC/mm^3^) [1]. The patient had no prolonged fever or meningeal symptoms during the fluid examination. Therefore, MOGAD aseptic meningitis was not considered.

Seizures occur in approximately 10.3% to 24.5% of patients with MOGAD [25,26,27]. They are more common in patients with cerebral cortical encephalitis (CCE) and ADEM-like phenotypes. Symptomatic seizures in MOGAD primarily happen during the first demyelinating episode (50%) or after it (27.8%), and they are least likely to occur before the episode (22.2%) [26,28]. Focal onset seizures are the most common type and can be associated with cortical lesions or show normal radiographic findings on MRI [26,28].

The exact mechanism behind seizures and epilepsy in MOGAD is not yet fully understood. However, it is likely more related to immune responses than to the presence of anti-MOG antibodies [25,26]. In MOGAD, the activation of CD4+ lymphocytes, infiltration of granulocytes, destruction of oligodendrocytes, demyelination, and gliosis occurs [26,29]. This demyelination process increases external potassium levels and heightened neuronal excitability [26,30]. Additionally, the release of proinflammatory cytokines and microglia activation may further influence excitatory and inhibitory effects within neuronal networks [26,29].

Epileptic seizures observed in the patient after the surgical treatment of the right adrenal tumor might have subsided after the treatment of the suspected paraneoplastic syndrome. At that time, serum anti-MOG antibodies were not measured. However, based on the literature review [9,25,26,31], it can be concluded that the seizures, specifically focal motor and secondary generalized seizures, were the initial manifestation of MOGAD. The diagnosis of ganglioneuroblastoma was discovered coincidentally and did not affect the subsequent diagnosis of MOGAD.

## 4. Conclusions

The patient’s case shows a multi-phase, clinically diverse course of MOGAD. Most importantly, it indicates the need to extend the diagnosis of epilepsy towards demyelinating diseases: the determination of anti-MOG and anti-AQP4 antibodies in the serum, especially in case of focal seizures.

## Figures and Tables

**Figure 1 neurolint-17-00037-f001:**
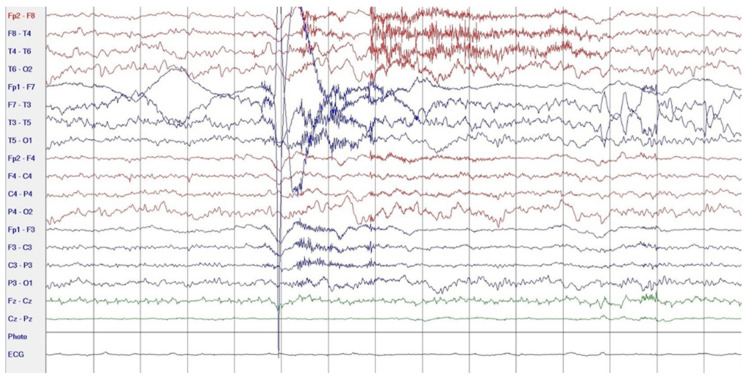
EEG trace: rapid intraictal rate (December 2020).

**Figure 2 neurolint-17-00037-f002:**
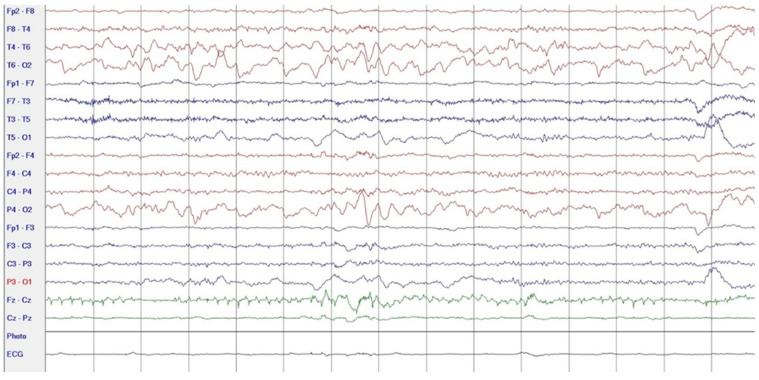
EEG recording: postictal theta waves (December 2020).

**Figure 3 neurolint-17-00037-f003:**
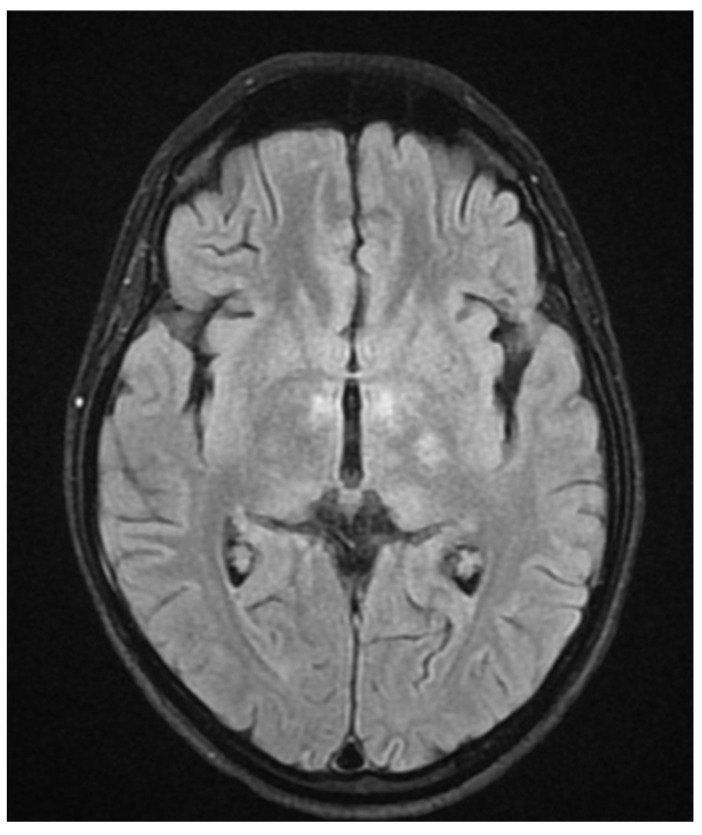
Transverse FLAIR brain magnetic resonance image showing numerous foci of increased signal between the basal nuclei (December 2021).

**Figure 4 neurolint-17-00037-f004:**
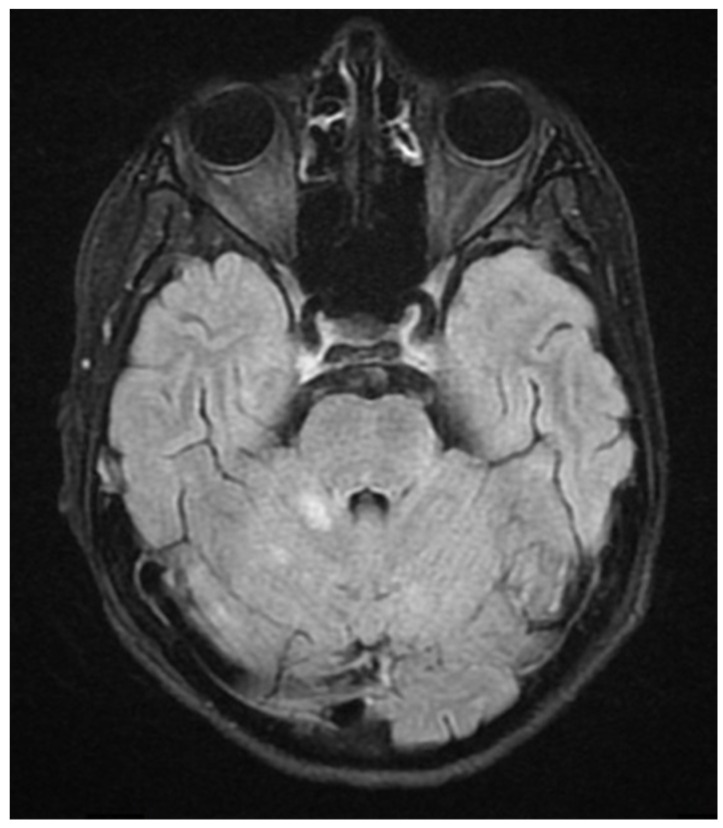
Transverse FLAIR brain magnetic resonance image showing area of increased signal in the right middle cerebellar peduncle (December 2021).

**Figure 5 neurolint-17-00037-f005:**
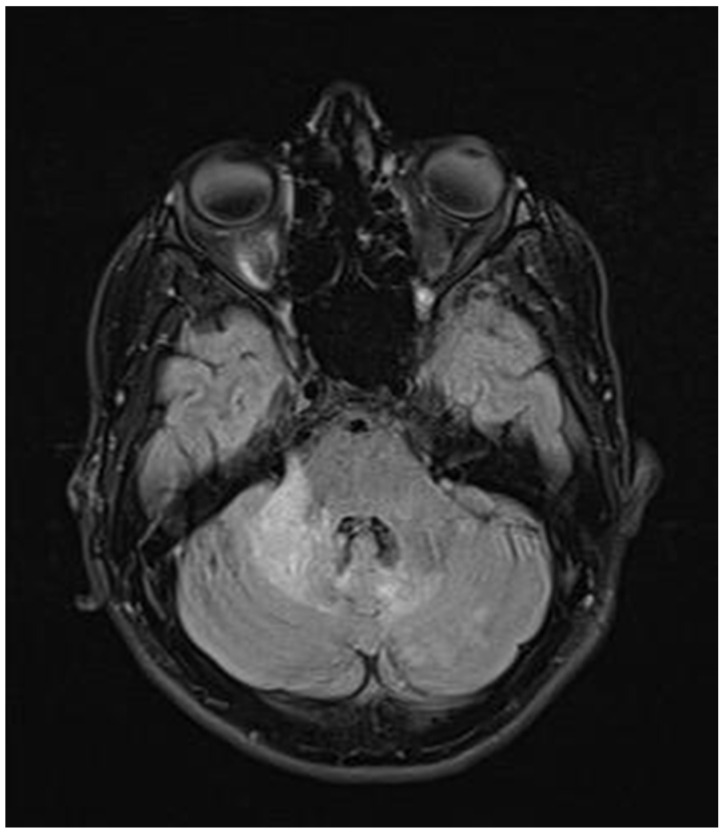
FLAIR image of brain magnetic resonance imaging in the transverse plane showing a demyelinating focus in the right hemisphere of the cerebellum, including its peduncle (March 2022).

**Figure 6 neurolint-17-00037-f006:**
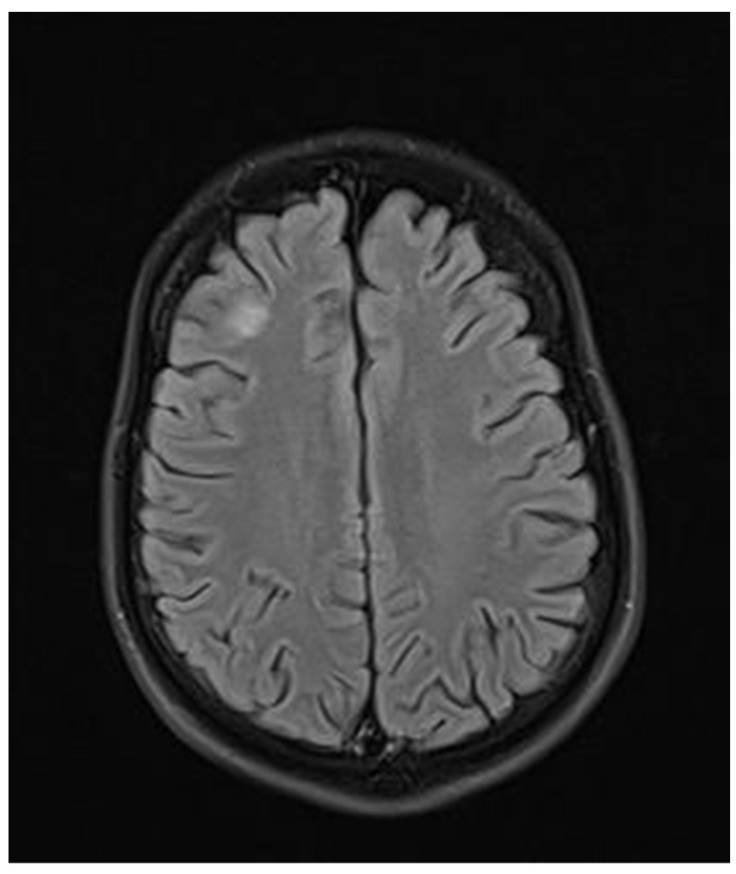
Transverse FLAIR brain magnetic resonance image showing a subcortical demyelinating focus in the right frontal lobe (March 2022).

**Figure 7 neurolint-17-00037-f007:**
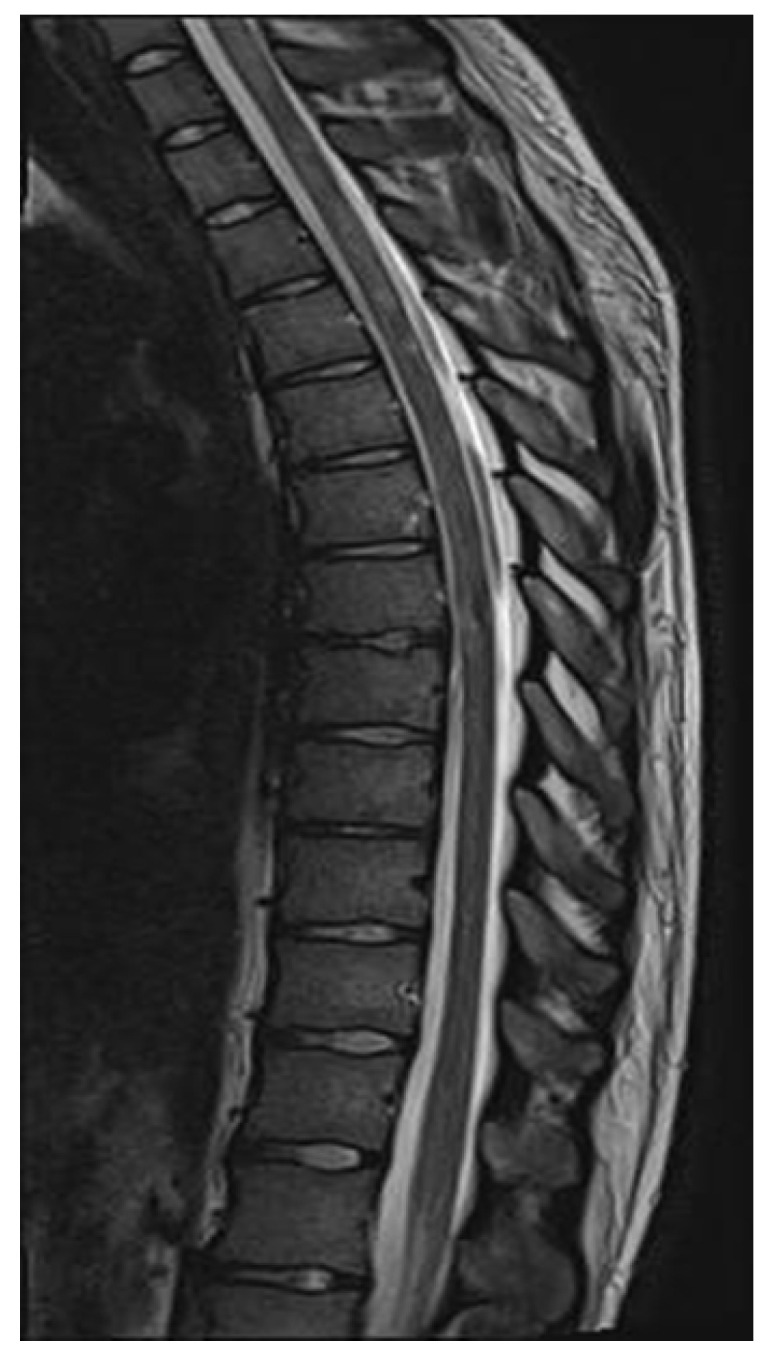
Spinal cord MRI showing foci of increased signals of thoracic and cervical spine on T2 sequences (March 2022).

**Figure 8 neurolint-17-00037-f008:**
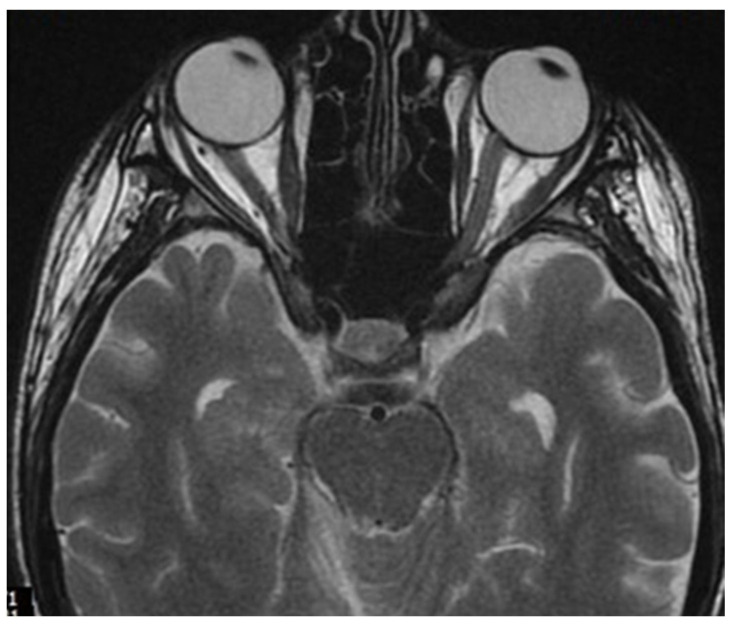
MRI of optic nerves showing a dilatation and edema of the proximal left optic nerve (March 2022).

**Figure 9 neurolint-17-00037-f009:**
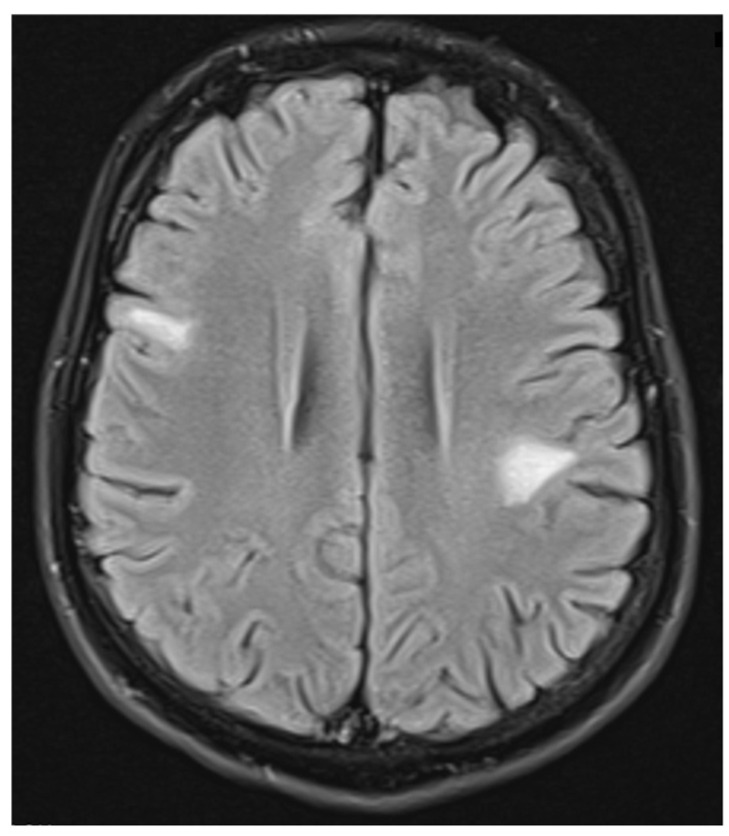
Transverse FLAIR brain magnetic resonance image showing two new subcortical foci of increased signals: one located in the right frontal lobe measuring 13 × 7 mm and another in the left parietal lobe measuring 19 × 14 mm. (May 2022).

**Figure 10 neurolint-17-00037-f010:**
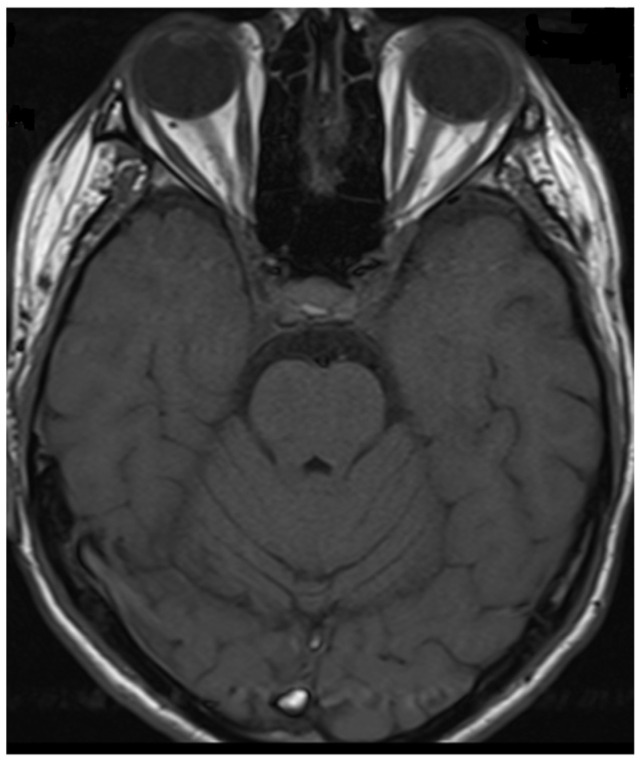
MRI of optic nerves showing the right optic nerve with blurred outlines (May 2022).

**Table 1 neurolint-17-00037-t001:** Demyelinating syndrome and supportive radiological features.

Demyelinating Syndrome	Supportive Radiological Features
Optic neuritis	bilateral simultaneous clinical involvement, longitudinal optic nerve involvement (>50% length of optic nerve), perineural optic sheath enhancement, optic disc edema
Myelitis	longitudinally extensive enhancement, central cord lesion, or H-sign, conus lesion
Brain, brainstem or cerebellar syndrome	multiple ill-defined T2-hyperintense lesions in supratentorial and often infratentorial white matter, deep gray matter involvement, ill-defined T2-hyperintensity involving pons, middle cerebellar peduncle, or medulla, cortical lesion with /without lesional and overlying meningeal enhancement

## Data Availability

No new data were created during the preparation of the case report.

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
