# Peer review of "Confusing Onset of MOGAD in the Form of Focal Seizures"

_2035-8377, 2025, doi:10.3390/neurolint17030037_

Round 1
Reviewer 1 Report
Comments and Suggestions for Authors
This case report presents an intriguing and rare instance of MOGAD with focal seizures preceding classic demyelinating features. The manuscript is well-structured and contributes to the growing understanding of MOGAD's heterogeneity. However, a few aspects require clarification and elaboration to improve the scientific rigor and clinical impact of the report.
- Authors Should draw a detailed timeline of disease progression and treatment approach.
- The authors state: "Brain MRI showed numerous changes in the increased signal on T2-weighted and FLAIR sequences: supratentorial, infratentorial, periventricular, and subcortical. The picture was found to be suggestive of ADEM. Inclusion of additional MRI images (specifically, more demonstrative ADEM examples) would strengthen the case. The manuscript mentions optic neuritis and spinal involvement, but corresponding imaging findings are not well detailed. Were any longitudinally extensive transverse myelitis (LETM) lesions seen? Did optic nerve MRI show enhancement, nerve sheath involvement, or chiasmal extension?
- The manuscript does not clearly state the MOG antibody titers at different stages.
- The case presentation describes left-sided optic neuritis, but the conclusion states:
"After the diagnosis of MOGAD with a typical radiological and clinical picture (ADEM, MDEM, binocular ON)..." Was optic neuritis initially unilateral and later bilateral, or was it bilateral from the start? The manuscript should clarify the evolution of optic neuritis and provide MRI confirmation for bilateral involvement. Given that the patient developed optic neuritis, was a VEP test conducted? VEP abnormalities could provide additional supportive evidence of demyelination and help in follow-up assessments. If VEP was not performed, this should be acknowledged as a limitation.
- The manuscript appropriately acknowledges the initial suspicion of paraneoplastic encephalitis given the history of ganglioneuroblastoma. However, the text should further emphasize that the co-occurrence of a tumor and MOGAD could be coincidental. MOGAD is a heterogeneous disease, and the role of anti-MOG antibodies in secondary autoimmunity remains uncertain. Paraneoplastic syndromes and MOGAD might have overlapping features but distinct mechanisms.
Author Response
For research article
|
Response to Reviewer X Comments
|
||
|
1. Summary |
|
|
|
Thank you very much for taking the time to review this manuscript. Please find the detailed responses below and the corresponding revisions/corrections highlighted/in track changes in the re-submitted files
|
||
|
2. Questions for General Evaluation |
Reviewer’s Evaluation |
Response and Revisions |
|
Does the introduction provide sufficient background and include all relevant references? |
Yes/Can be improved/Must be improved/Not applicable |
|
|
Are all the cited references relevant to the research? |
Yes/Can be improved/Must be improved/Not applicable |
|
|
Is the research design appropriate? |
Yes/Can be improved/Must be improved/Not applicable |
|
|
Are the methods adequately described? |
Yes/Can be improved/Must be improved/Not applicable |
|
|
Are the results clearly presented? |
Yes/Can be improved/Must be improved/Not applicable |
|
|
Are the conclusions supported by the results? |
Yes/Can be improved/Must be improved/Not applicable |
|
|
3. Point-by-point response to Comments and Suggestions for Authors
|
||
|
Comments 1: Authors Should draw a detailed timeline of disease progression and treatment approach. The authors state: "Brain MRI showed numerous changes in the increased signal on T2-weighted and FLAIR sequences: supratentorial, infratentorial, periventricular, and subcortical. The picture was found to be suggestive of ADEM. Inclusion of additional MRI images (specifically, more demonstrative ADEM examples) would strengthen the case. The manuscript mentions optic neuritis and spinal involvement, but corresponding imaging findings are not well detailed. Were any longitudinally extensive transverse myelitis (LETM) lesions seen? Did optic nerve MRI show enhancement, nerve sheath involvement, or chiasmal extension?
|
||
|
Response 1: Thank you for paying attention. We agree with this comment. Therefore, we have expanded the description of head MRIs performed in December 2021 (line 71 to 78), and we added images from this period (fig. 3, fig. 4) also the description of the MRI from March 2022 was supplemented with a clear statement that these were LETM-type changes (lines 96 to 99) and an image from this MRI was added (fig. 7). Also, the description of the orbital resonance has been corrected (lines 99 to 101) and enriched with an illustration (fig.8)
|
||
|
Comments 2: The manuscript does not clearly state the MOG antibody titers at different stages. |
||
|
Response 2: We agree. Therefore, we added the anti-MOG antibody titer from December 2021 (line 82) and from March 2022 (line 102)
Comments 3: " The case presentation describes left-sided optic neuritis, but the conclusion states: "After the diagnosis of MOGAD with a typical radiological and clinical picture (ADEM, MDEM, binocular ON)..." Was optic neuritis initially unilateral and later bilateral, or was it bilateral from the start? The manuscript should clarify the evolution of optic neuritis and provide MRI confirmation for bilateral involvement. Given that the patient developed optic neuritis, was a VEP test conducted? VEP abnormalities could provide additional supportive evidence of demyelination and help in follow-up assessments. If VEP was not performed, this should be acknowledged as a limitation.”
Response 3: We agree that it was not clearly described. We have expanded the description of the episode of recurrent optic neuritis (lines 106 to 120) after left-sided inflammation in March 2022 and right-sided inflammation occurred in May 2022. The description has been enriched with images (fig.9 and Fig. 10). Instead of "bilateral," there should be "recurrent" optic neuritis (corrected in lines 26, 118, and 123). Visual potentials were not performed, but OCT RNFL and MRI of the head and orbits were performed, which confirmed optic neuritis.
Comments 4: The manuscript appropriately acknowledges the initial suspicion of paraneoplastic encephalitis given the history of ganglioneuroblastoma. However, the text should further emphasize that the co-occurrence of a tumor and MOGAD could be coincidental. MOGAD is a heterogeneous disease, and the role of anti-MOG antibodies in secondary autoimmunity remains uncertain. Paraneoplastic syndromes and MOGAD might have overlapping features but distinct mechanisms.
Response 4: We agree and have expanded the discussion to include the potential pathomechanism of epileptic seizures (lines 209 to 216). While emphasizing that the diagnosis of ganglioneuroblastoma was a coincidence and had no impact on the occurrence of MOGAD (lines 220 to 223)
|
||
Reviewer 2 Report
Comments and Suggestions for Authors The authors describe a patient that started out as epilepsy and later turned out to be MOGAD due to the appearance of central nervous system lesions. Although this is a valuable case report that provides insight into epilepsy as a rare form of MOGAD, the authors need to reconsider the following points. What are the possible mechanisms that caused the epilepsy in this patient? Aseptic meningitis has been reported as a rare form of MOGAD (https://doi.org/10.1212/WNL.0000000000205621). The authors mentioned that there was no abnormality in the CSF at the onset of epilepsy in this patient, but were there any other findings suggestive of aseptic meningitis? Line 117 resonance imaging → magnetic resonance imaging Line 118 magnetic resonance imaging → MRIAuthor Response
For research article
|
Response to Reviewer X Comments
|
||
|
1. Summary |
|
|
|
Thank you very much for taking the time to review this manuscript. Please find the detailed responses below and the corresponding revisions/corrections highlighted/in track changes in the re-submitted files
|
||
|
2. Questions for General Evaluation |
Reviewer’s Evaluation |
Response and Revisions |
|
Does the introduction provide sufficient background and include all relevant references? |
Yes/Can be improved/Must be improved/Not applicable |
|
|
Are all the cited references relevant to the research? |
Yes/Can be improved/Must be improved/Not applicable |
|
|
Is the research design appropriate? |
Yes/Can be improved/Must be improved/Not applicable |
|
|
Are the methods adequately described? |
Yes/Can be improved/Must be improved/Not applicable |
|
|
Are the results clearly presented? |
Yes/Can be improved/Must be improved/Not applicable |
|
|
Are the conclusions supported by the results? |
Yes/Can be improved/Must be improved/Not applicable |
|
|
3. Point-by-point response to Comments and Suggestions for Authors
|
||
|
Comments 1: What are the possible mechanisms that caused the epilepsy in this patient? Aseptic meningitis has been reported as a rare form of MOGAD (https://doi.org/10.1212/WNL.0000000000205621). |
||
|
Response 1: Thank you for paying attention. We agree and have expanded the discussion to include the potential pathomechanism of epileptic seizures (lines 209 to 216). Aseptic meningitis has indeed been reported as a rare form of MOGAD. We have reviewed the literature and added this phenotype to the introduction (lines 45 to 48). However, our patient was never diagnosed with aseptic meningitis - during the cerebrospinal fluid examination, he had no fever (especially for more than 7 days) or meningeal symptoms (lines 200-202).
|
||
|
Comments 2: The authors mentioned that there was no abnormality in the CSF at the onset of epilepsy in this patient, but were there any other findings suggestive of aseptic meningitis? Line 117 resonance imaging → magnetic resonance imaging Line 118 magnetic resonance imaging → MRI
Response 2: Thank you for paying attention. However, after analyzing the literature (references 11-16) on aseptic meningitis in MOGAD and considering our patient's clinical picture, we could not diagnose aseptic meningitis. |
||
Reviewer 3 Report
Comments and Suggestions for Authors
Dears,
The manuscript "Confusing onset of MOGAD in the form of focal seizures" is an interesting case report. It describes a case of a 17-year-old patient presenting with focal seizures, that was finally diagnosed with MOGAD.
There are some mistakes:
- line 58 - in the sentence "four generalized tonic-clonic seizures were also observed" "episodes" before "of tonic-clonic seizures" should be added
- line 62 - instead of "acute... complexes" rather "sharp"
- line 73 - the titer of anti-MOG antibodies should be given in number, if it was assessed
- line 144 - there is a mistake: in 15-30 % of MOGAD patients, oligoclonal bands are positive, not anti-MOG antibodies.
The article may be published after corrections.
Best greetings,
Reviewer
Comments on the Quality of English Language
There are some English language mistakes that require correction by a native speaker.
Author Response
For research article
|
Response to Reviewer X Comments
|
||
|
1. Summary |
|
|
|
Thank you very much for taking the time to review this manuscript. Please find the detailed responses below and the corresponding revisions/corrections highlighted/in track changes in the re-submitted files
|
||
|
2. Questions for General Evaluation |
Reviewer’s Evaluation |
Response and Revisions |
|
Does the introduction provide sufficient background and include all relevant references? |
Yes/Can be improved/Must be improved/Not applicable |
|
|
Are all the cited references relevant to the research? |
Yes/Can be improved/Must be improved/Not applicable |
|
|
Is the research design appropriate? |
Yes/Can be improved/Must be improved/Not applicable |
|
|
Are the methods adequately described? |
Yes/Can be improved/Must be improved/Not applicable |
|
|
Are the results clearly presented? |
Yes/Can be improved/Must be improved/Not applicable |
|
|
Are the conclusions supported by the results? |
Yes/Can be improved/Must be improved/Not applicable |
|
|
3. Point-by-point response to Comments and Suggestions for Authors
|
||
|
Comments 1: There are some mistakes:
line 58 - in the sentence "four generalized tonic-clonic seizures were also observed" "episodes" before "of tonic-clonic seizures" should be added
|
||
|
Response 1: Thank you for paying attention. We agree with this comment. Therefore, we have corrected line 61-62 “However, four generalized episodes of tonic-clonic seizures were also observed during treatment.”
|
||
|
Comments 2: There are some mistakes: line 62 - instead of "acute... complexes" rather "sharp" |
||
|
Response 2: Thank you for paying attention. We agree with this comment. Therefore, we have corrected line 64-65 “Interictal EEG showed rapid activity followed by slow theta waves and several sharp and slow wave complexes”
Comments 3: There are some mistakes: line 73 - the titer of anti-MOG antibodies should be given in number, if it was assessed
Response 3: We agree. Therefore, we added the anti-MOG antibody titer from December 2021 (line 82) and from March 2022 (line 102)
Comments 4: There are some mistakes: line 144 - there is a mistake: in 15-30 % of MOGAD patients, oligoclonal bands are positive, not anti-MOG antibodies.
Response 4: We agree with this comment. Therefore, we have corrected line 198 “in 15-30% of MOGAD patients, oligoclonal bands are positive”
4. Response to Comments on the Quality of English Language Point 1: “There are some English language mistakes that require correction by a native speaker.” Response 1: We agree; therefore, the proofread article has been rechecked by a native speaker.
|
||